# A Thin-Film Pinned-Photodiode Imager Pixel with Fully Monolithic Fabrication and beyond 1Me- Full Well Capacity

**DOI:** 10.3390/s23218803

**Published:** 2023-10-29

**Authors:** Joo Hyoung Kim, Francois Berghmans, Abu Bakar Siddik, Irem Sutcu, Isabel Pintor Monroy, Jehyeok Yu, Tristan Weydts, Epimitheas Georgitzikis, Jubin Kang, Yannick Baines, Yannick Hermans, Naresh Chandrasekaran, Florian De Roose, Griet Uytterhoeven, Renaud Puybaret, Yunlong Li, Itai Lieberman, Gauri Karve, David Cheyns, Jan Genoe, Paweł E. Malinowski, Paul Heremans, Kris Myny, Nikolas Papadopoulos, Jiwon Lee

**Affiliations:** 1Imec, Kapeldreef 75, 3001 Leuven, Belgium; joo.hyoung.kim@imec.be (J.H.K.); francois.berghmans@imec.be (F.B.); abu.bakar.siddik@imec.be (A.B.S.); irem.sutcu@imec.be (I.S.); isabel.pintormonroy@imec.be (I.P.M.); jehyeok.yu.ext@imec.be (J.Y.); tristan.weydts@imec.be (T.W.); epimitheas.georgitzikis@imec.be (E.G.); jubin.kang.ext@imec.be (J.K.); yannick.baines@imec.be (Y.B.); yannick.hermans@imec.be (Y.H.); naresh.chandrasekaran@imec.be (N.C.); florian.deroose@imec.be (F.D.R.); griet.uytterhoeven@imec.be (G.U.); renaud.puybaret@imec.be (R.P.); yunlong.li@imec.be (Y.L.); itai.lieberman@imec.be (I.L.); gauri.karve@imec.be (G.K.); david.cheyns@imec.be (D.C.); jan.genoe@imec.be (J.G.); pawel.malinowski@imec.be (P.E.M.); paul.heremans@imec.be (P.H.); kris.myny@imec.be (K.M.); nikolas.papadopoulos@imec.be (N.P.); 2Department of Electrical Engineering (ESAT), KU Leuven, 3001 Leuven, Belgium; 3College of Information and Communication Engineering, SKKU, Suwon 16419, Republic of Korea; 4Department of Electrical Engineering, Ulsan National Institute of Science and Technology (UNIST), Ulsan 44919, Republic of Korea; 5Department of Photonics and Nanoelectronics and the BK21 FOUR ERICA-ACE Center, Hanyang University ERICA, Ansan 15495, Republic of Korea

**Keywords:** thin-film photodiode, large full well capacity, high dynamic range

## Abstract

Thin-film photodiodes (TFPD) monolithically integrated on the Si Read-Out Integrated Circuitry (ROIC) are promising imaging platforms when beyond-silicon optoelectronic properties are required. Although TFPD device performance has improved significantly, the pixel development has been limited in terms of noise characteristics compared to the Si-based image sensors. Here, a thin-film-based pinned photodiode (TF-PPD) structure is presented, showing reduced kTC noise and dark current, accompanied with a high conversion gain (CG). Indium-gallium-zinc oxide (IGZO) thin-film transistors and quantum dot photodiodes are integrated sequentially on the Si ROIC in a fully monolithic scheme with the introduction of photogate (PG) to achieve PPD operation. This PG brings not only a low noise performance, but also a high full well capacity (FWC) coming from the large capacitance of its metal-oxide-semiconductor (MOS). Hence, the FWC of the pixel is boosted up to 1.37 Me- with a 5 μm pixel pitch, which is 8.3 times larger than the FWC that the TFPD junction capacitor can store. This large FWC, along with the inherent low noise characteristics of the TF-PPD, leads to the three-digit dynamic range (DR) of 100.2 dB. Unlike a Si-based PG pixel, dark current contribution from the depleted semiconductor interfaces is limited, thanks to the wide energy band gap of the IGZO channel material used in this work. We expect that this novel 4 T pixel architecture can accelerate the deployment of monolithic TFPD imaging technology, as it has worked for CMOS Image sensors (CIS).

## 1. Introduction

Monolithically processed thin-film photodiodes (TFPDs) on the Si ROIC (Read-Out Integrated Circuitry) are attractive imaging platforms when beyond-silicon optoelectronic properties are required [1,2]. They can sense photons with energy smaller than the Si bandgap (1.12 eV) or can absorb light more efficiently leading to a much reduced active layer thickness while enabling small form factor and higher resolution visible imaging [1,3,4,5,6,7]. Despite these appealing features, the signal-to-noise ratio (SNR) of the TFPD imager pixel output is limited by the kTC noise, high dark current, and low Conversion Gain (CG), compared to the conventional Complementary Metal Oxide Semiconductor (CMOS) Image sensor (CIS). For the Si-based image sensors, a pinned photodiode (PPD) pixel architecture was introduced to address the issues above [8,9,10]. It was designed to single out the image lag and the kTC noise by pinning the maximum potential of the integration node when it is reset. The photoelectrons are collected in the photodiode during the integration time and these signal charges are transferred to the floating diffusion (FD) to suppress the dark current and elevate the CG. To implement the Si PPD operation, thin-film pinned photodiode (TF-PPD) pixel architecture was demonstrated with the typical low-noise readout operation [11]. The TF-PPD structure is realized by inserting a thin-film transistor (TFT) module based on an oxide semiconductor (here, indium-gallium-zinc oxide: IGZO) between the Si ROIC and the TFPD (Figure 1). The photogate (PG) pins the PD reset level, while the dark current from the depleted surface is suppressed by the wide energy band gap semiconductor (IGZO, Figure 2f). The transfer gate (TG) enables the charge integration and transfer operation. In this way, proof-of-concept TF-PPD has been successfully implemented by mimicking operation of the Si PPD [11]. In this paper, a large full well capacity (FWC) exceeding 1 Mega electrons of the proposed pixel (pitch 5 μm) is highlighted, relying on the fact that PG is a MOS capacitor, which serves as an integration node (Figure 1). This feature could not be well addressed in the previous report, due to the limited FD capacitance of the active pixel [11]. It is found that the high-k material boosts its FWC as a gate dielectric. Dark currents and their activation energies are compared between the PD test structure and TF-PPD passive pixel, to figure out the leakage contribution from the depleted semiconductor interface, inherent to the PG structure, that was not well covered in our previous report [11].

## 2. Device Configuration and Measurements

The TF-PPD pixel is built in a fully monolithic scheme on a custom-made Si ROIC (Figure 1b). First, the Si ROIC is fabricated with the 130 nm CMOS process exposing PG, TG, and FD electrodes on the top surface. A 10 nm Al_2_O_3_ gate dielectric is deposited via atomic layer deposition followed by sputtering of 12 nm thick IGZO. TG/PG electrodes, Al_2_O_3_ and IGZO layers constitute a back-gated TFT structure, while the FD connects the TFT to the Si ROIC. The colloidal quantum dot (CQD) light-absorbing layer, hole transport layer, and the top transparent contact (ITO: indium tin oxide) are layered sequentially, with the IGZO TFT channel simultaneously acting as an electron transport layer (ETL) at the bottom. Details of device fabrication can be found in our previous report [11]. The dark and photocurrent characteristics of the PD test structure are given in Figure 2a. The IGZO is chosen due to its low leakage, high mobility, and compatibility with the CQD PD as an ETL [11,12]. The fabricated IGZO TFT shows I-V characteristics with an on/off ratio larger than 10^5^ and a V_th_ of −2 V (Figure 2b).

Various designs of passive TF-PPD pixels, each with more than 500 parallel-connected arrays, are fabricated as described above and then characterized with a custom-made printed circuit board (PCB) probe card (Figure 2c,d). The PCB is integrated with an off-the-shelf CTIA (Texas Instruments ACF2101) for the signal charge-to-voltage conversion and the field-programmable gate array (Xilinx Artix-7) for generation of control signals. The output of the CTIA is recorded using a DSOX3014 oscilloscope. The TF-PPD pixels are illuminated using a ThorLabs M530L4 530 nm LED, which is modulated via a ThorLabs DC2200 LED driver. Integration capacitance for the CTIA is increased by adding discrete capacitors after the capacitances are measured using a HM8118 LCR bridge to handle the large FWC of the TF-PPD pixels.

## 3. Simulation and Experimental Results

### 3.1. TCAD Simulations

Device simulations are performed using Silvaco TCAD to verify the single-pixel TF-PPD operation. The physical structure of the TF-PPD is defined in the Athena process simulator. The structure file is then introduced to the Silvaco TCAD Atlas numerical device simulator to figure out electrical characteristics. Prior to the integration of the a-IGZO transfer gate onto the pixel, experimental results of the TFT electrical characteristics are reflected by adopting a sub bandgap density of states (cm^−3^ eV^−1^) model within the reported bandgap [13]. The material properties of the TFT such as work function, electron affinity, optical bandgap, and the properties of the photodiode are adopted from our experimental results and Silvaco organic library, respectively. The electrostatic potential is simulated for the a-IGZO layer all along the PG, TG, and FD as can be seen in Figure 3d. The TFT channel exhibits a favorable potential profile for the full charge transfer as expected for the TF-PPD pixel architecture (Figure 3b). The potential level of the integration node (PG) is pinned higher than the TG on value, while that is lower than the TG off value. Signal charge integration and charge transfer phase can be well distinguished by turning on and off the TG, showing that the TF-PPD is realizable with the introduction of the PG structure (Figure 3c,d). More rigorous simulation is our ongoing research work, to understand TF-PPD pixel operation better.

### 3.2. Pixel Output Responses by the PG and TG Bias Sweep

By changing the PG or TG bias, the signal charge generation, and the transfer of it can be controlled (Figure 4). First, with the fixed TG on a bias of −1 V, V_PG_ is swept from −4 V to −1 V, where the FD reset voltage is set to 0 V by CTIA, and the TG off value is −6.5 V. When V_PG_ is set to −4 V, which is the same bias as V_anode_, meaning a limited reverse bias of the PD, a suppressed signal output is found (Figure 4a). As the V_PG_ increases, implying a larger reverse bias within PD, more photocurrent conducts, which is expected from Figure 2a and Figure 4a. In other words, by controlling the V_PG_, the sensitivity of the PD can be controlled.

V_TG_ sweep measurements are carried out with the fixed V_PG_ of −2 V. The V_TG_ is scanned from −6.5 V to −1 V, showing the limited signal output when the V_TG_ is −6.5 V (Figure 4b). According to the TFT I-V curve given in Figure 2b, the TG begins to be turned on after −2 V. As the VTG increases, more charge transfer becomes available, showing a larger photocurrent for a higher VTG (Figure 4b). However, substantial charge transfer is observed before V_TG_ −2 V, which can be explained by the V_th_ nonuniformity within the passive pixel array.

### 3.3. Full-Well Capacity (FWC)

The PPD concept offers inherent low noise, where the recent PPD-based CIS pixels have a few electrons of read noise level [14,15]. In addition, the active area of the TF-PPD pixel is defined by the PG, thus generated signal charges are collected at the MOS capacitor, rather than the PD junction capacitor [16]. For the proposed pixel architecture, the FWC is described by the capacitor equation of Q = CV, where C = εA/d (Q: charge stored at the MOS capacitor; V: PD voltage swing; ε: dielectric permittivity; A: capacitor area; d: gate dielectric thickness). By choosing high-k material as the gate dielectric and thinning the layer, the FWC of the TF-PPD can be boosted. Thanks to this PG structure, the FWC is estimated to be up to 1.1 Me- with the 5 μm pixel pitch and 81% fill factor, followed by 1.4 Me- of measured FWC (Table 1 and Figure 6b, 1.5 V of PD voltage swing assumed by simulation, ε_r_ = 9 for AlOx [17,18]). This is more than eight times larger than the estimated PD junction capacitance, with the same PD voltage swing and 100% fill factor to avoid underestimation of it. As the pixel pitch increases from 5 μm to 10 μm, the elevation of the FWC can be observed, reaching up to 4 Me- for a 10 μm pixel with the fill factor of 50%, while 5.6 Me- of the FWC is measured for the pi-shaped pixel (fill factor 81% (Table 1, Figure 5b and Figure 6b)). This observation implies that the generated signal charges are well collected at the MOS capacitor of the TFT channel on the integration node (Figure 3). As a result, a charge transfer happens along the IGZO channel, not through the PD, so the charge transport mainly depends on TFT properties rather than TFPD, which usually has lower mobility than Si. 

In addition, the FWC increase due to the fill factor is studied, which also shows a boosted FWC for a higher fill factor (Figure 6). This confirms that the FWC of the pixel is well described by the MOSCAP of the IGZO channel, meaning that the PG is working as expected.

Finally, the FWC is measured by changing the V_PG_ (Figure 7). With the larger voltage swing, a further increase in FWC is expected, accompanied by more photocurrent with a higher PD reverse bias (Figure 7). Once again, an increase in the FWC can be found for the VPG change, and in turn, the PD voltage swing (Figure 7b). Here, an IGZO potential that is 0.5 V lower compared to the V_PG_ is assumed for the FWC estimation, due to the work function difference between the TiN electrode and the IGZO [11,19].

### 3.4. Dark Current

One of the benefits of the PPD scheme is the surface passivation and the separation of the depletion regions from the interface, while the photogate pixel does not guarantee these configurations [16]. Here, since the channel is made with the IGZO, which has a wider bandgap than Si (3.5 vs. 1.12 eV), the leakage current from the depleted interface is expected to be smaller than in the Si case (Figure 2f). To confirm this, temperature-dependent dark current measurements are carried out (Figure 8b) for the passive pixel and PD test structure. The measured dark current showed similar values (Figure 2a and Figure 8). From the Arrhenius plot, activation energies are found close to each other, 0.40 eV for the QDPD test structure (1420 nm peak absorption) and 0.39 eV for the passive pixel, implying that they would share the same dark current source. For the PD test structure, the IGZO acts just as an ETL, and the dark current contribution by the depletion of this layer is not considered major [20]. Extracted activation values are well aligned with the previous report, showing almost half the energy of the full bandgap of QDs tuned for the same wavelength absorption, which can be interpreted that most of dark charges are generated from the PD or its interfaces, rather than the depleted IGZO/Al_2_O_3_ interface of PG [21].

## 4. Discussion

This beyond -1Me- FWC within a 5 µm pixel pitch shows one of the largest signal storage capabilities, showing a high FWC density of 55 ke-/µm^2^ (Figure 9, Table 2). Though this value is already high enough, it can be even further boosted by having higher-k materials and by the thinning the gate dielectric. Pixels with higher FWC density per unit area can be found from the literature; however, most of them have adopted additional capacitors such as the lateral overflow integration trench capacitor [22], deep trench isolation with hole collection [23]. 

DRAM capacitor [40], Metal–Insulator–Metal capacitor [41], or 3-dimensional capacitor [42]. Assuming low readout noise as presented in our previous report [11], with the high FWC presented in this work, a high linear dynamic range over 100 dB is estimated, while the 3 T pixel shows 82 dB, presenting a significant increase in the dynamic range by the introduction of the novel pixel architecture [11,21]. Realizing this large dynamic range for the active pixel array is our ongoing research work.

Since the un-covered FD area is sensitive to the light, this provides the parasitic light signal (Figure 1b). However, considering the typical operation of 4 T image sensors, where the time interval between the sampling reset level and the signal level is less than a few μs (with the integration time of ~ms), this parasitic effect would be limited, except for the extremely high illumination conditions. Despite this, it would still be advantageous to cover the FD by a light shielding structure and focus the light onto the integration node (here, the PG area), e.g., using micro lenses. 

## 5. Conclusions

In this work, a TF-PPD pixel is proposed with the full monolithic scheme from Si ROIC to TFPD, with the insertion of a IGZO TFT module between them. The PPD scheme inherently provides low noise, and by the introduction of the PG, the FWC is boosted up to 1.4 Me- with the 5 µm pixel pitch. This beyond-megaelectron FWC can be well modeled by the MOS capacitor of the PG, demonstrating that generated signal charges at the PD are subsequently collected at the TFT channel, and then transferred along this layer. From these results, it is expected that the presented TF-PPD pixel will serve as a high SNR pixel topology for the TFPD category of image sensors, which can properly complement imaging applications that are not or poorly covered by the Si imagers.

## Figures and Tables

**Figure 1 sensors-23-08803-f001:**
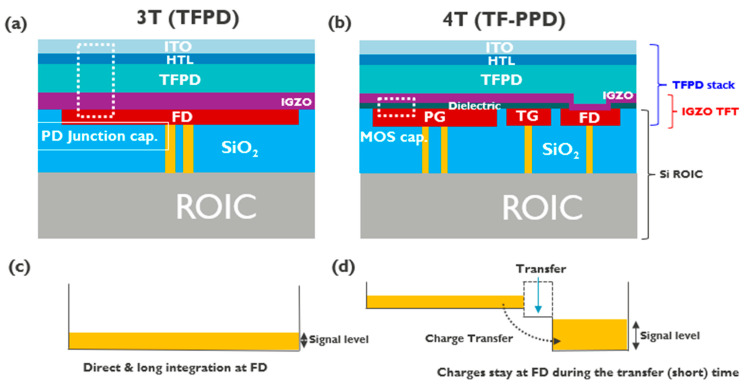
Pixel cross-section for the monolithic TFPD image sensor (**a**) 3 T and (**b**) 4 T (TF-PPD) structure (TCO: transparent conductive oxide, HTL: hole transport layer, PG: photogate, TG: transfer gate, FD: floating diffusion). Electric potential and signal readout configuration for 3 T pixel (**c**) and for 4 T pixel (**d**). Pixel circuit diagram for 3 T pixel (**e**) and for the 4 T pixel (**f**).

**Figure 2 sensors-23-08803-f002:**
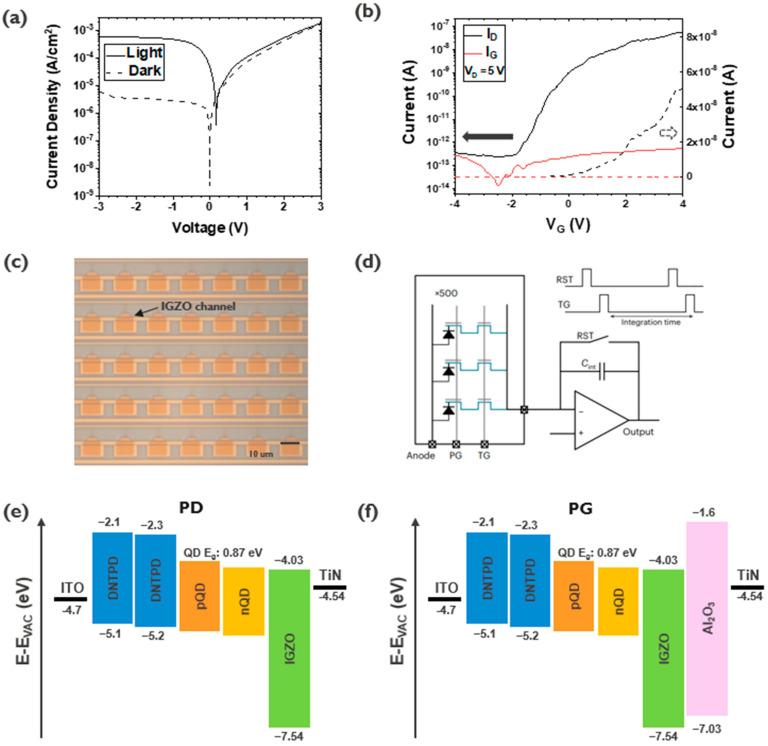
I-V characteristic of QDPD test structure (**a**) and of IGZO TFT (**b**), a micrograph of the TF-PPD passive pixel array (**c**), and its measurement schematic (**d**). Band diagrams for the PD (**e**) and PG (**f**).

**Figure 3 sensors-23-08803-f003:**
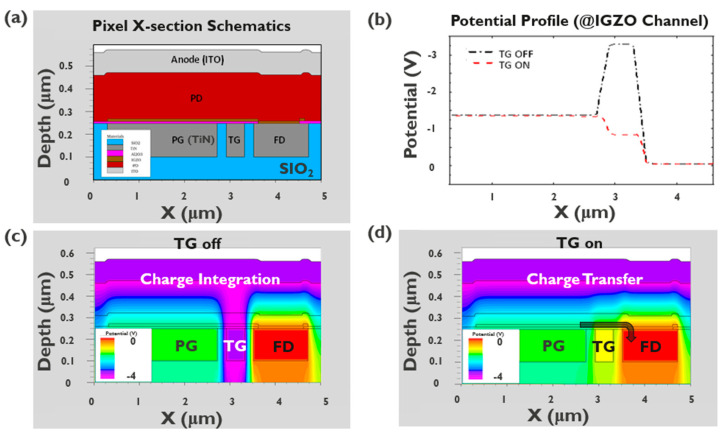
Silvaco TCAD simulation results; (**a**) simulated structure, (**b**) lateral potential profile along the IGZO layer, and (**c**) potential profile when TG is turned off and (**d**) on.

**Figure 4 sensors-23-08803-f004:**
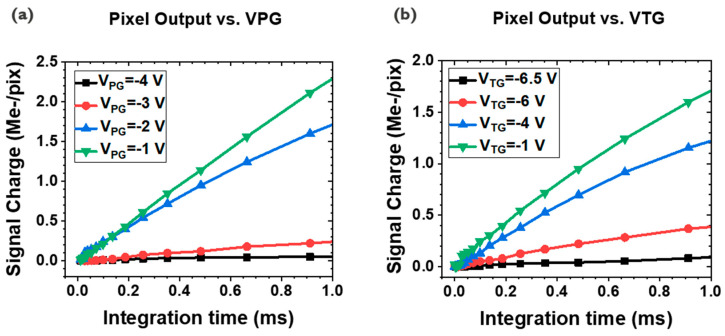
Signal output vs. integration time with different VPG and VTG values with the illumination. Signal curves with the fixed VTG (−1 V), varying VPG (−4~−1 V) (**a**), the same graphs for the fixed VPG (−2 V), and different VTGs (−6.5~−1 V) (**b**).

**Figure 5 sensors-23-08803-f005:**
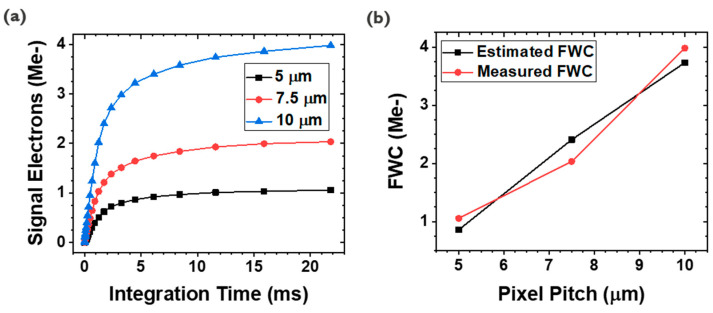
(**a**) Pixel output vs. integration time for different pixel pitches. (**b**) FWC comparison between estimation and measurement.

**Figure 6 sensors-23-08803-f006:**
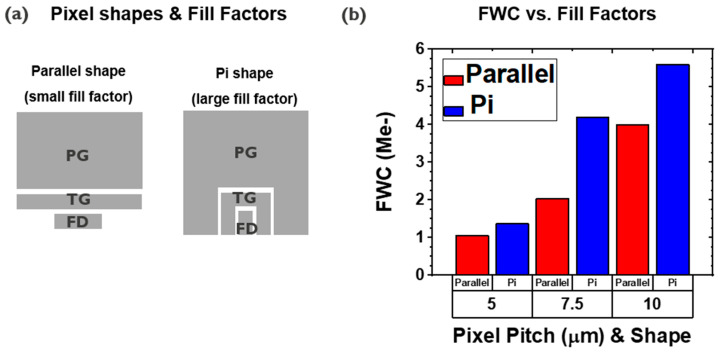
FWC comparison by different pixel fill factors. Pixel schematics for different shapes (**a**), and FWC by different pixel shapes and pitches (**b**).

**Figure 7 sensors-23-08803-f007:**
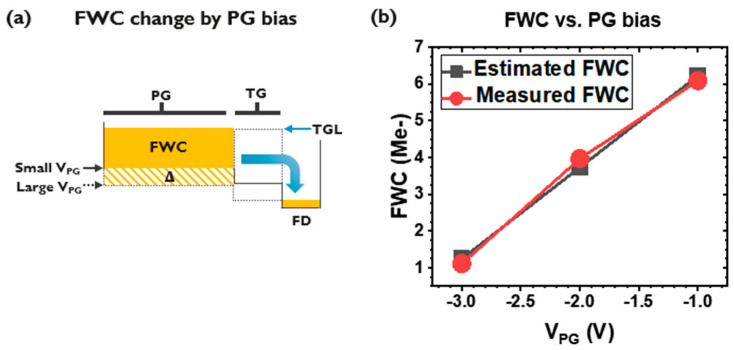
Potential diagram describing FWC increase by the larger VPG (**a**), and FWC vs. V_PG_ (**b**).

**Figure 8 sensors-23-08803-f008:**
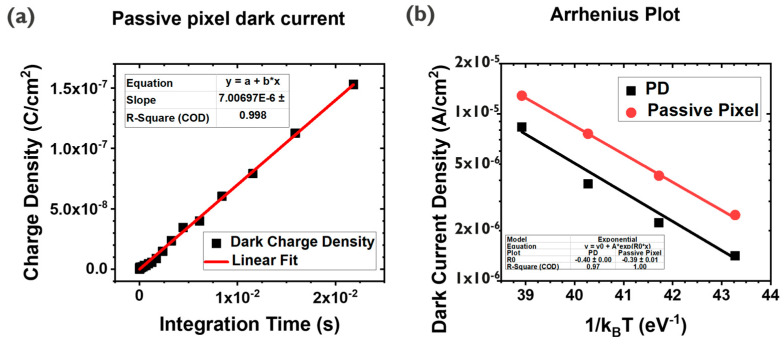
Passive pixel dark current (**a**) and Arrhenius plots (**b**) for the QDPD test structure and the passive pixel.

**Figure 9 sensors-23-08803-f009:**
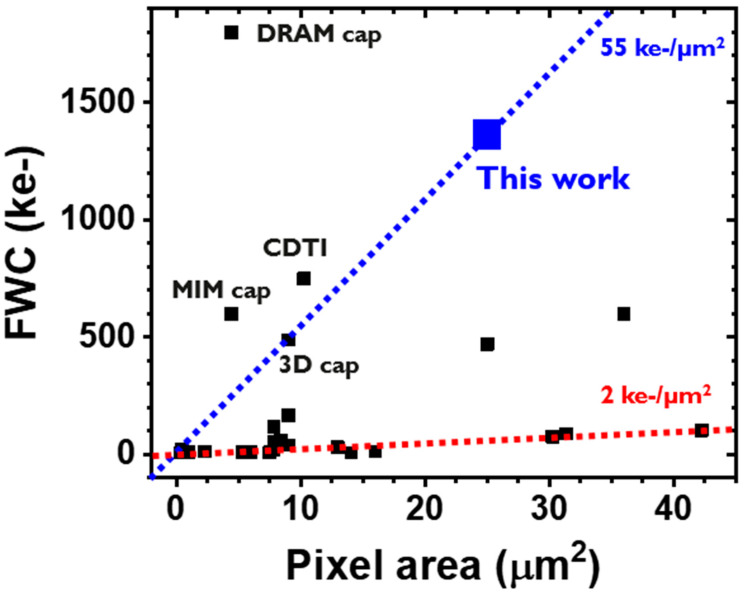
FWC vs. pixel area. A guideline showing the FWC density per unit area for this work (blue) and a trend line for the most of CISs (red).

**Table 1 sensors-23-08803-t001:** Estimated and measured FWC of TF-PPD pixels.

Pixel Pitch	Fill Factor	Estimated FWC	Measured FWC
**5 μm**	46%	0.9 Me-	1.1 Me-
	73%	1.4 Me-	1.4 Me-
**7.5 μm**	57%	2.4 Me-	2.0 Me-
	88%	3.7 Me-	4.2 Me-
**10 μm**	50%	3.7 Me-	4.0 Me-
	81%	6.1 Me-	5.6 Me-

**Table 2 sensors-23-08803-t002:** FWC comparison for different pixels.

Ref.	FWC (ke-)	Pixel Pitch (µm)	FWC Density (ke-/µm^2^)	Remarks
This work	1367	5	55	TF-PPD
[22]	24,300	16	95	LOFIT Trench cap
[23]	750	3.2	73	CDTI, hole collection
[24]	103	6.5	2	-
[25]	12	1	12	-
[26]	20	2.8	3	-
[27]	10	0.6	28	-
[28]	8	2.4	1	-
[29]	120	2.8	15	-
[30]	20	0.64	49	-
[31]	76	5.5	3	-
[32]	13	1.5	6	-
[33]	55	2.8	7	-
[34]	10	2.74	1	-
[35]	88	5.6	3	-
[36]	30	3.6	2	-
[37]	12	2.3	2	-
[38]	40	3	4	-
[39]	60	2.9	7	-
[40]	1800	2.1	408	DRAM cap
[41]	600	2.1	136	MIM cap
[42]	166	3	18	Dual PD
[5]	470	5	19	TFPD Junction cap
[43]	10	2.45	2	-
[44]	489	3	54	3D cap
[45]	8	3.75	1	-
[46]	600	6	17	TFPD Junction cap

## Data Availability

The data used to support the findings of this study are available from the corresponding author upon request.

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
