# Peer review of "A Thin-Film Pinned-Photodiode Imager Pixel with Fully Monolithic Fabrication and beyond 1Me- Full Well Capacity"

_sensors, 2023, doi:10.3390/s23218803_

Round 1

Reviewer 1 Report

Comments and Suggestions for Authors

What is the difference between the submitted paper (A Thin-Film Pinned-Photodiode Imager Pixel with Fully Monolithic Fabrication and beyond 1Me- Full Well Capacity) and the one published in Nature Electronics (Thin-film image sensors with a pinned photodiode structure)?

An extensive explanation must be given mainly on similarities and differences between both studies.

Author Response

We do not wish to claim novelties of the current paper in every respect, but rather to note that the current paper is an extension of the previous one. In this paper, the low full well capacity and the dark current from depleted interface, which were considered as potential drawbacks in our previous report, are extensively analyzed.

The Nature paper focused on that how this pixel can be built from the photodiode level up to the image sensor demonstration, with the special emphasis on its low noise performance and general pixel operations. For this paper, large FWC of our pixel is highlighted, coming from its photogate structure, which has MOS capacitor, boosting the FWC beyond 1 Mega electron within 5 um pixel pitch (Section 3.3). It is found that this FWC is one of the highest according to our literature survey (Section 4, Fig.9 added to highlight this). This feature cannot be well addressed in the previous report, due to the limited Floating Diffusion capacitance. However, in this paper, using passive pixel, that limitation can be lifted, and maximum attainable signal level from the integration node is properly measured (1. Introduction). This indicates that our pixel not just lowers the noise, but also improves maximum attainable signal, enabling 3-digit dynamic range in the end (Section 4.). Moreover, activation energy of the dark current is studied to see if there is substantial contribution from the depleted interface of charge transfer channel and the gate dielectric (Section 3.4). This has been one of the failure reasons with the Si photogate pixel, while for our case, by the introduction of large band gap channel material, the limited impact is found in terms of dark current and its activation energy. In addition, TCAD simulation part provides details of pixel operation (Section 3.1).

We agree that further elaboration on the application of the current technology would benefit the reader's understanding, so the paper is now updated as recommended.

Location of change in revised manuscript

Page 2, Line 58, 59, 60-62, 64, 65, 69

Page 3, Line 83, 84

Page 8, Line 207

Page 8, Fig. 9

Page 9, Table 2

Page 11, Ref. 12

Page 12, Ref. 30-45

Reviewer 2 Report

Comments and Suggestions for Authors

In this paper (Communication), the Authors present the latest results of technological research on monolithic optical high-dynamic-range sensors based on thin-film photodiodes. The article is written concisely and reads well. However, I have a few questions/comments. What is the potential use of such imagers? Will they replace classic CMOS image sensors in the future? It is worth writing a few words about it in the introduction or conclusions.

Can the pixel nonlinearity (Figure 4) be given in some standard, e.g. INL?

Do you plan to implement a larger pixel array with CDS?

Thank you.

Author Response

Comment 1

Reviewer wrote:

In this paper (Communication), the Authors present the latest results of technological research on monolithic optical high-dynamic-range sensors based on thin-film photodiodes. The article is written concisely and reads well. However, I have a few questions/comments. What is the potential use of such imagers? Will they replace classic CMOS image sensors in the future? It is worth writing a few words about it in the introduction or conclusions.

Our response: Due to the versatility of the proposed pixel structure, the current pixel can be used in general thin film-based image sensing applications, such as extending the wavelength to near infrared, short infrared, and x-ray, while enabling small form factor and higher resolution visible imaging (1. Introduction).

We’d like to bring this technology to complement where the Si imager cannot or poorly cover with the by-par pixel performance (5. Conclusions).

We agree that further elaboration on the application of the current technology would benefit the reader's understanding, so the paper is now updated as recommended.

Location of change in revised manuscript

Page 1, Line 40

Page 10, Line 239, 240

Comment 2

Reviewer wrote:

Can the pixel nonlinearity (Figure 4) be given in some standard, e.g. INL?

Our response:

EMVA standard recommends that the linearity error of the image sensors is calculated by computing deviation from the ideal linear curve which is defined by a least-squares linear regression minimizing the relative deviation of the 10% to 90% of the output values. The measured linearity error of the image sensor can be found in [10]. In Fig. 4 we wanted to show that the output sensitivity is modulated by changing the potential applied to PG and TG and comparing the linearity is not the main intention. We agree that the term 'linearity' in Fig 4 can be misleading, so it is now updated as 'output' to avoid confusion.

Location of change in revised manuscript

Page 5, Top captions of Fig. 4

Page 5, Title of the Section 3.2

Comment 3

Reviewer wrote:

Do you plan to implement a larger pixel array with CDS?

Our response:

It is pity that the image sensor with maximum use of the full well capacity has not been implemented in the previous design [10]. We are working on the new design to fully exploit the capabilities of the thin film PPD pixel (added in 4. Discussions). However, we believe that the current pixel-level demonstration of large full-well capacity without increasing dark current by using the MOS capacitance of the photogate structure with large bandgap channel material is beneficial for readers.

Location of change in revised manuscript

Page 9, Line 222, 223

Reviewer 3 Report

Comments and Suggestions for Authors

Author Response

Comment 1

Reviewer wrote:

[Improvements from Ref.[10]] The large FWC is clearly demonstrated. However, what has changed from the previous report (Ref. [10]) is unclear. Is it due to PG structure ? or layout ? or else ? Since the PG and PPD device structure seem to be identical to those of Ref. [10], please clarify this point.

Our response:

We do not wish to claim that the current paper presents a new pixel design, but please understand that it presents the extended analysis. One of the weakness of the previous paper [10] was that the total FWC is limited by the FD capacity, so an accurate assessment of the FWC was not possible We would like to clarify this point and show that even large FWC is attainable with the proposed pixel structure, by measuring the same passive pixels presented in [10] with larger integration capacitance (Section 2. Device Configuration and Measurements, Fig. 2(d)). By doing so, we believe that the current presentation would be informative and potentially beneficial to readers by broadening their understanding of the proposed pixel architecture.

We now revised the paper according to your comment, this point is once again emphasized considering more intuitive understanding by the general readers.

Location of change in revised manuscript

Page 2, Line 64, 65

Comment 2

Reviewer wrote:

[MOS capacitor design] To complement the above point, please give a pixel circuit diagram including the PG-based MOS capacitor corresponding to Figures 1 (a) and (b).

Our response:

Figure 1 is updated reflecting reviewer’s comment.

Location of change in revised manuscript

Page 2, Fig. 1 (e), (f).

Comment 3

Reviewer wrote:

[Term of “PPD”] I’m confused with the terminology of “PPD” in the following ways. (1) In Si PPDs, fluctuation of the surface potential is shielded (“pinned”) by high density hole accumulation layer leading to suppression of dark current through (near-)surface states. Although briefly discussed in section 3.4, a description of such aspect seems to be insufficient. Please add more detailed description on the origins of the dark current and on how they are suppressed by the present scheme leading to the identical term of “PPD”. Does IGZO TFT automatically give pinning like hole accumulation layer in Si CIS ? Probably, explanation with a band diagram would be helpful. (2) In general, we expect complete charge transfer and complete depletion of PD after a reset operation for conventional PPDs. Please clarify how complete these are in the present device.

Our response:

(1) We agree that the term 'pinned' was originally used to describe surface pinning only, as the reviewer commented, but we believe that it is now also widely used to describe the maximum potential pinning of the PPD. As an example, we could find about 230 results after searching for the keyword ''pinning voltage' image sensor'' from google scholar search and we kindly ask you to refer to some of the representative results below. In our proposed pixel structure, the maximum potential of the photodiode is pinned by using photogate biasing, while the dark current from the depleted surface is suppressed by using a wide energy band gap semiconductor (IGZO, EG = 3.51 eV). In this way, we would argue that we have successfully mimicked the operation of the PPD and that is why we have used the PPD for our works.

From your comment we noticed that this point was not clearly presented in the paper, so it is now updated as suggested (3.4. Dark Current, Fig. 2 (e), (f) for the band diagrams).

We have tried to list some of literatures that can support how much reasonable is the term “PPD” for our proposed pixel topology below:

[R2.1] Marcelot, Olivier, et al. "Pinned photodiode CMOS image sensor TCAD simulation: In-depth analysis of in-pixel pinning voltage measurement for a diagnostic tool." IEEE transactions on electron devices 64.2 (2016): 455-462.

[R2.2] Goiffon, Vincent, et al. "Radiation effects in pinned photodiode CMOS image sensors: Pixel performance degradation due to total ionizing dose." IEEE Transactions on Nuclear Science 59.6 (2012): 2878-2887.

[R2.3] Tan, Jiaming, Bernhard Buttgen, and Albert JP Theuwissen. "Analyzing the radiation degradation of 4-transistor deep submicron technology CMOS image sensors." IEEE Sensors Journal 12.6 (2012): 2278-2286.

[R2.4] Kim, Seong-Jin. "Performance evaluation of pinning potential adjustment in two-dimensional/three-dimensional image sensor." IEEE electron device letters 33.10 (2012): 1426-1428.

[R2.5] Goto, Yotaro, et al. "Pinning Voltage Control of CMOS Image Sensor by Measuring Sheet Resistance at Micro Test Structure in Scribe Line." 2017 17th International Workshop on Junction Technology (IWJT). IEEE, 2017.

(2) The charge transfer performance of the proposed pixels has been presented in [10] and we kindly ask you to refer to it for further information. There were 3.5% and 1.5% of charges remaining after transfer for 10µm and 7.5µm pixels respectively. This might be attributed to the spacing between PG to TG and FD (200nm each), where the electrostatic potentials are less tightly controlled. Further detailed analysis is currently being carried out to figure out why this less perfect charge transfer happens.

Location of change in revised manuscript

Page 2, Line 60

Page 3, Figure 2(e), (f)

Page 7, Line 189, 197-199

Comment 4

Reviewer wrote:

[Light shielding in the FD area] There seems to be no light shielding in the FD area. Doesn’t this affect SNR during the integration/reset period? If so, by how much? Please add some comments on this point.

Our response:

The unshielded FD area will certainly respond to light and this parasitic photocurrent would contribute to the signal level. This FD light sensitivity would degrade the light signal, by not proportional to the integration time, but to the light intensity. However, for the typical operation of the image sensors, where the interval between the sampling reset and the signal, is less than a few µs (typical integration time of ~ms), this parasitic effect is limited, except for the extremely high illumination conditions. Therefore, it would be advantageous to cover the FD by using a light shielding structure and focus the light onto the integration node (here PG area), e. g. by a micro lens. This argument is added to the Section 4. Discussion.

Location of change in revised manuscript

Page 10, Line 224-230

Comment 5

Reviewer wrote:

[Figure 2 (b)] Please clearly indicate which curve belongs to which (right or left) vertical axis just like the format used in Fig. 2h of Ref. [10] (by arrows and circles).

Our response:

Arrows are added as the reviewer has suggested in Fig. 2(b).

Location of change in revised manuscript

Page 3, Fig. 2(b)

Comment 6

Reviewer wrote:

[Figure 8 (b)] Please give the real dark current values in the vertical axis rather than the logarithmic function. It would be much easier for readers to grasp the good level of the device.

Our response:

The figure is updated as suggested by the reviewer.

Location of change in revised manuscript

Page 8, Fig. 8(b)

Round 2

Reviewer 3 Report

Comments and Suggestions for Authors

For most of the previous comments, the authors thoroughly and properly responded. I appreciate the authors’ significant effort.

 One final comment is made regarding to the following statement;

 p.1 Line 45-p.2 Line 51

It was designed to single out the kTC noise by pinning the maximum potential of the integration node when it is reset.”

 It was, I think, originally developed to reduce image lag according to [8]. So, I recommend replacing the above “kTC noise” with “image lag”.

Author Response

Dear reviewer,

It is indeed true that the PPD is developed to reduce the image lag. I have added this to the sentence reflecting your comment (p.2 line 51). At the same time, by the same scheme that the PPD has suppressed the image lag, it can be seen that the kTC noise can be elliminated. This discussion can be also found in the PPD review paper (Fossum, Eric R. and Hondongwa, Donald B., "A Review of the Pinned Photodiode for CCD and CMOS Image Sensors" (2014). Dartmouth Scholarship. 2423. https://digitalcommons.dartmouth.edu/facoa/2423, p.34, , 1st column, line 30~35, 2nd column, line 29-31), which is added as the ref. [10] for the better understanding by the reader.

Best,

Joo Hyoung
